# Temporal Trend of the SARS-CoV-2 Omicron Variant and RSV in the Nasal Cavity and Accuracy of the Newly Developed Antigen-Detecting Rapid Diagnostic Test

**DOI:** 10.3390/diagnostics14010119

**Published:** 2024-01-04

**Authors:** Daisuke Tamura, Yuji Morisawa, Takashi Mato, Shin Nunomiya, Masaki Yoshihiro, Yuta Maehara, Shizuka Ito, Yasushi Ochiai, Hirokazu Yamagishi, Toshihiro Tajima, Takanori Yamagata, Hitoshi Osaka

**Affiliations:** 1Department of Pediatrics, Jichi Medical University, 3311-1 Yakushiji, Shimotsuke 329-0498, Japan; r0954hy@jichi.ac.jp (H.Y.);; 2Department of Infectious Disease, Jichi Medical University, 3311-1 Yakushiji, Shimotsuke 329-0498, Japan; 3Department of Emergency Center, Jichi Medical University, 3311-1 Yakushiji, Shimotsuke 329-0498, Japan; 4Department of Intensive Care Unit, Jichi Medical University, 3311-1 Yakushiji, Shimotsuke 329-0498, Japan; 5Research & Development Division, Sekisui Medical Co., Ltd., Chuo-ku, Tokyo 103-0027, Japan

**Keywords:** omicron, rapid diagnostic test, RT-PCR, RSV, SARS-CoV-2, viral antigens

## Abstract

The aim of this work is to analyze the viral titers of severe acute respiratory syndrome coronavirus 2 (SARS-CoV-2) and respiratory syncytial virus (RSV) at the anterior nasal site (ANS) and nasopharyngeal site (NS), evaluate their virological dynamics, and validate the usefulness of a newly developed two-antigen-detecting rapid antigen diagnostic test (Ag-RDT) that simultaneously detects SARS-CoV-2 and RSV using clinical specimens. This study included 195 asymptomatic to severely ill patients. Overall, 668 specimens were collected simultaneously from the ANS and NS. The cycle threshold (Ct) values calculated from real-time polymerase chain reaction were used to analyze temporal changes in viral load and evaluate the sensitivity and specificity of the Ag-RDT. The mean Ct values for SARS-CoV-2-positive, ANS, and NS specimens were 28.8, 28.9, and 28.7, respectively. The mean Ct values for RSV-positive, ANS, and NS specimens were 28.7, 28.8, and 28.6, respectively. SARS-CoV-2 and RSV showed the same trend in viral load, although the viral load of NS was higher than that of ANS. The sensitivity and specificity of the newly developed Ag-RDT were excellent in specimens collected up to 10 days after the onset of SARS-CoV-2 infection and up to 6 days after the onset of RSV infection.

## 1. Introduction

Since the emergence of severe acute respiratory syndrome coronavirus 2 (SARS-CoV-2) in Wuhan, China in December 2019, various new variants have been continually identified worldwide. Vaccinations have been administered since 2020 to control new outbreaks, although the number of symptomatic infections continues to increase [1]. The Omicron variant, which emerged in 2020, has caused a pandemic in its short history, being transmitted worldwide at an unprecedented speed and creating new cases of infection. The Omicron variant places a heavy burden on healthcare facilities owing to its high infectivity and high rate of serious illnesses [2]. One of the main reasons for the high infectivity of Omicron is the presence of numerous mutations and deletions within the gene encoding the spike protein [3,4], which increases the ability of the virus to escape liquid immunity and strongly bind to the receptor angiotensin-converting enzyme [5]. Thus, the Omicron variant is more adaptable and proliferates rapidly in the human body [6,7]. Compared to the Delta variant, the viral load is known to be higher in the internal part of the nasal cavity, especially the anterior nasal cavity [8]. However, no studies have analyzed and evaluated the viral dynamics of the Omicron variant at different sites in the nasal cavity over time or in different age groups. Respiratory syncytial virus (RSV), similar to other respiratory viruses such as parainfluenza virus and human metapneumovirus, causes acute upper and lower respiratory tract infections (bronchitis, bronchiolitis, and pneumonia). Notably, in infants, transferable maternal antibodies are ineffective in protecting against infection, and high rates of bronchitis and pneumonia are associated with respiratory failure at the time of initial infection. Approximately half of infants infected with RSV have lower respiratory tract infections, and 3% of all infants are hospitalized [9]. Lower respiratory tract infections caused by RSV in older individuals are known to be complicated by wheezing in 6–30% of cases [10]. Furthermore, RSV re-infection exacerbates chronic diseases in older patients with underlying diseases such as chronic respiratory diseases (asthma and chronic obstructive pulmonary disease). Therefore, early diagnosis and prevention measures are important in homes, daycare centers, and hospitals to avoid transmission to infants and older individuals from those with mild cases of infection.

Rapid and accurate diagnostic tests for SARS-CoV-2 are essential in any medical strategy to combat coronavirus disease-2019 (COVID-19). Tests used to diagnose SARS-CoV-2 include: (1) detection of viral RNA by nucleic acid amplification tests (NAAT) such as real-time reverse transcription polymerase chain reaction (RT-PCR) and (2) detection of viral antigens by immunodiagnostic techniques such as lateral flow assays, commonly referred to as antigen detection rapid diagnostic tests (Ag-RDTs) [11,12]. Clinical laboratories worldwide have performed over three billion molecular diagnostic tests for SARS-CoV-2 [13]. The need for highly accurate NAAT for SARS-CoV-2 testing is indisputable. However, the limited number of staff members with skills and knowledge and the high cost of testing clearly make it difficult to implement NAAT on a continuous and sustainable basis worldwide [14]. In contrast, Ag-RDTs contribute to pathogen diagnosis and treatment without the need for training or special laboratory procedures, thus, reducing reliance on laboratory infrastructure [15]. Since the COVID-19 pandemic, hundreds of Ag-RDTs have been commercialized and clinically implemented [16]. Many clinical specimens have been approved for use, including nasopharyngeal and pharyngeal site specimens, followed by saliva, nasopharyngeal aspirate, and sputum, although those used for self-testing are often limited to saliva or anterior nasal site (ANS) specimens [17,18]. Studies on the detection efficacy of SARS-CoV-2 variants for Ag-RDT are divided [19,20]. As with SARS-CoV-2, RSV diagnosis can be made using PCR, cell culture, and Ag-RDT antigen detection using enzyme antibodies or immunochromatography; however, the sensitivity of Ag-RDT for detecting RSV antigen is generally approximately 70–80%, and more accurate diagnostic methods are needed [21].

Ag-RDTs are superior in their portability, that is, in their ability to be performed anywhere, and rapidity, that is, they produce results within 20–30 min from the start of the test. However, they are not as versatile as multiplex RT-PCR, and only one specimen can be used with one Ag-RDT. Therefore, the practical application of Ag-RDTs to help diagnose COVID-19 and other diseases with similar clinical symptoms, including RSV infection, from a single specimen is eagerly awaited in clinical practice during the COVID-19 pandemic.

Although the viral load of the Delta variant in ANS specimens is lower than that in nasopharyngeal site (NS) specimens, the former has proven useful for testing. Therefore, ANS specimens, which are medically safe, have been used for self-testing. However, for the Omicron variant, without a careful comparison of viral levels in the NS and ANS, it is not possible to determine whether ANS specimens are suitable for self-testing. The World Health Organization (WHO) suggests that new mutant variants of SARS-CoV-2 may lower the accuracy of Ag-RDT testing and recommends that the specificity and sensitivity of Ag-RDT be evaluated for each mutant variant [22]. The primary site of growth of RSV is the epithelial cells of the lower respiratory tract [23], and NS is the most common site for the collection of viral antigen specimens for diagnosis; however, to date, the utility of ANS specimens for RSV diagnosis has not been studied or clarified.

The objective of this study was to determine the nasal viral dynamics of two different viruses, the Omicron variant of SARS-CoV-2 and RSV, by analyzing the viral load and viral shedding course over time at the two nasal sites: ANS and NS. Viral titers at the ANS were used to determine whether the ANS could be clinically applicable for COVID-19 self-testing and whether the ANS could serve as an alternative specimen collection site for RSV. The second objective was to validate the diagnostic ability of the newly developed Ag-RD, which is a combination of separate Ag-RDTs already in use, to identify SARS-CoV-2 and RSV antigens within a single Ag-RDT. The basic principle of the detection of SARS-CoV-2 and RSV antigens is based on existing technologies with several technical improvements. The diagnostic accuracy of the newly developed Ag-RDT was evaluated using clinical specimens because its diagnostic accuracy was unclear.

## 2. Materials and Methods

### 2.1. Participants

This prospective cross-sectional study was conducted at the Jichi Children’s Medical Center Tochigi and the Intensive Care Unit and Emergency Center of Jichi Medical University Hospital, Japan, from 1 February 2022, to 31 July 2023. During this period, a major outbreak of COVID-19 occurred in Japan, mainly due to the Omicron variant [24]. Sporadic outbreaks of RSV have also been reported in kindergartens, elementary schools, and elderly care facilities.

Patients who were considered as outpatients or hospitalized for suspected SARS-CoV-2 or RSV infections were included. Patients who had had intense contact with COVID-19 patients, regardless of the presence or absence of symptoms, were also included. Eligible patients were tested for the presence of SARS-CoV-2 or RSV antigens independently of this study using RT-PCR in the hospital laboratory while the attending physician performed various examinations during routine medical care. The exclusion criteria for this study were as follows: (i) high risk of bleeding from the nasal site due to concurrent bleeding disorders; (ii) inability or unwillingness of the patients or their guardians to provide written informed consent; (iii) date of the first specimen collection exceeding 10 days after disease onset; and (ⅳ) more than 10 days had passed since close contact with a patient with COVID-19 (Figure 1). If patients confirmed that they did not meet any of the exclusion criteria and provided written consent, they were invited to participate in the study. A patient was considered to have mild disease when they did not require oxygen inhalation and maintained an oxygen saturation ≥ 93% and a patient was considered to have moderate to severe disease when their oxygen saturation was <93% with findings of pneumonia and requirement of oxygen supplementation or ventilatory support. Children hospitalized for encephalopathy, seizures, or myocarditis were diagnosed with moderate or severe disease, even if their respiratory symptoms were mild. All experiments involving human participants were performed in accordance with the Declaration of Helsinki, and all participants provided written informed consent. For participants younger than 18 years, written consent was obtained from a surrogate parent or guardian. This study was approved by the Research Ethics Review Committee of Jichi Medical University Hospital (approval no. 21-100).

### 2.2. Newly Developed Ag-RDT Measurement Principle

The newly developed RapidTesta RSV and SARS-CoV-2 (Sekisui Medical Co., Ltd., Tokyo, Japan) is an Ag-RDT that can simultaneously detect two viral antigens, SARS-CoV-2 and RSV, in a single specimen (Appendix A). The detection technology for each viral antigen is based on Rapid Testa SARS-CoV-2 (Sekisui Medical Co., Ltd., Tokyo, Japan) for SARS-CoV-2 antigen detection and RapidTesta RSV-Adeno NEXT (Sekisui Medical Co., Ltd., Tokyo, Japan) for RSV and adenoviral antigen detection, both of which are already widely used in clinical practice; both require ≤10 min for the final determination by visual judgement or RapidTesta Reader. Analysis by the RapidTesta Reader was used to measure the absorbance values of antigen lines; if a clear antigen-positive line was visually determined, analysis by the RapidTesta Reader was deemed unnecessary. Evaluation data for the respective Ag-RDTs show that RapidTesta SARS-CoV-2 has a sensitivity of 89.5% and specificity of 100% [25], whereas RapidTesta RSV-Adeno NEXT shows a sensitivity of 98.1% and specificity of 100% for RSV antigen detection [26].

### 2.3. Testing Procedures

We prospectively enrolled individuals with suspected SARS-CoV-2 or RSV infection because of symptoms of malaise, anorexia, and respiratory distress, along with a fever of ≥37.5 °C. We also prospectively enrolled non-symptomatic staff and their family members who had been determined to be close contact after an infection outbreak at the hospital. Records of clinical symptoms were obtained from all patients from whom clinical specimens were obtained.

Specimens were collected during outpatient visits. When multiple specimens were collected, in the case of inpatients, they were collected before meals or before nurses or physical therapists provided oral care or rehabilitation so as not to interfere with treatment and recuperation. In the case of outpatients, specimens were collected in an isolation room after determining in advance about their next visit to the hospital. Specimens from the ANS and NS were collected as previously reported [25]. Four specimens were collected simultaneously, two each from the ANS and NS. Of the two specimens collected from each site, one was evaluated using the newly developed RapidTesta RSV and SARS-CoV-2 test and the other was evaluated using the comparison RapidTesta RSV-Adeno NEXT or Rapid Testa SARS-CoV-2 kit that had already been selected prior to specimen collection. Randomization was used for the selection of RapidTesta RSV-Adeno NEXT or Rapid Testa SARS-CoV-2 as the comparator.

All specimens were collected by a single physician to minimize variation due to specimen collection techniques. Ag-RDT analysis was performed in a safety cabinet (biosafety level 2) within 1 h after specimen collection from the patient, according to the manufacturer’s instructions [27]. Specimens were dipped into the buffer solution supplied with the Ag-RDT and dropped into the small window of the Ag-RDT, and the results were visually read after the waiting period. The results were evaluated using RapidTesta Reader after visual evaluation, as shown in Appendix A, and considered definitive. After the Ag-RDT test, the residual buffer was stored at −80 °C within 1 h and was used for RT-PCR, which is discussed later. RNA extraction was performed using the QIAamp Viral RNA Mini Kit (Qiagen, Hilden, Germany). RT-PCR using primers/probes targeting two different regions of SARS-CoV-2 was performed according to the National Institute of Infectious Disease protocols using the Reliance One-Step Multiplex Supermix (Bio-Rad Laboratories, Hercules, CA, USA) [28]. Simultaneously, a two-plex RT-PCR reaction using probes differentiating RSV-A and RSV-B was performed according to the Chiba City Environmental Health Research Institute protocols using the One Step PrimeScript RT-PCR Kit and the manufacturer’s instructions (Takara Korea Biomedical Inc., Shiga, Japan) [29]. Even with RT-PCR, a low viral load may result in a false-negative result, decreasing the accuracy of the test. Therefore, it is important to evaluate the cycle threshold (Ct value) for the viral load [30]. The Ct value for the detection limit of Ag-RDT was determined based on the average value of the false-negative samples. Mutation analysis and determination of the variant using KANEKA RT-PCR Kit SARS-CoV-2 (Omicron/Delta) ver.2 (Kaneka Co., Ltd., Tokyo, Japan) was performed according to the manufacturer’s instructions.

### 2.4. Statistical Analyses

The sensitivity and specificity of the RapidTesta RSV and SARS-CoV-2 test were calculated for both ANS and NS specimens separately, using the RT-PCR results for comparison. The R software program version 4.1.0 (www.r-project.org, accessed on 2 December 2023) (R Foundation for Statistical Computing, Vienna, Austria) was used for statistical analysis. The Clopper and Pearson method was used to calculate the sensitivity and specificity of the Ag-RDT and the 95% confidence interval. Kappa values, positive predictive value (PPV), and negative predictive value (NPV) were also calculated to verify the concordance of the results.

## 3. Results

### 3.1. Participants

A total of 195 (97 pediatric and 98 adult) patients were included in this study, and their ages ranged from 0 months to 79 years. The mean and median ages were 24.1 and 16.0 years, respectively. Of the 195 patients, 114 (64 adults and 50 children) participated in the SARS-CoV-2 study, and 81 (34 adults and 47 children) participated in the RSV study (Figure 1). Overall, 668 specimens were collected from patients and analyzed, including 334 specimens each from the ANS, which is near the nostrils, and NS, which is the deep nasal site. The clinical determinations of COVID-19 and RSV infections in those who tested positive for viral antigens by RT-PCR are shown in Table 1. A total of 4 adult patients and 19 pediatric patients (6 with COVID-19 and 13 with RSV infection) were hospitalized for moderate to severe disease. All four adult patients were on ventilator management owing to COVID-19-related worsening respiratory status and were treated with remdesivir (200 mg intravenous (i.v.) on the first day and 100 mg i.v. on the second and subsequent days) and dexamethasone (6 mg i.v. once daily for 4–6 days). Of the six pediatric patients admitted with COVID-19, one had acute encephalopathy and another had viral pneumonia and required oxygen supplementation. The other four patients had febrile convulsions, croup syndrome, and moderate dehydration. Remdesivir was used for treatment but steroids were not. Monoclonal antibody therapies, such as casiribimab and imdevimab, were not used in all patients because of their low efficacy against the Omicron variant [31]. None of the adult patient required hospitalization for RSV infections. Thirteen pediatric patients were hospitalized for RSV infection: nine with bronchiolitis, three with viral pneumonia, and one with febrile convulsion. The asymptomatic patients who participated owing to their close contact with COVID-19 patients included 19 adults and 1 pediatric patient.

All adult patients had received the third through fifth doses of COVID-19 vaccine at least three months before specimen collection. In the pediatric patients, four had completed the initial vaccination (first and second doses) schedule.

### 3.2. SARS-CoV-2 Positivity

Of the 668 specimens, 162 ANS and 171 NS specimens tested positive for SARS-CoV-2 by RT-PCR. In the subtype analysis of the viral pathogens, all specimens were the Omicron variant. The mean Ct values for SARS-CoV-2 with positive RT-PCR results and for ANS and NS specimens were 28.8 (range: 16.9–40.9), 28.9 (range: 18.2–40.9), and 28.7 (range: 16.9–40.5), respectively (Figure 2). Moreover, among the positive RT-PCR specimens, 95 ANS specimens (58.6%) had higher Ct values than the NS specimens. The Ct values for ANS and NS specimens remained parallel over time. Comparison of the Ct values for both site specimens by day of onset showed that the viral load in NS specimens was high on most days, except on days 7, 9, 13, and 16 of onset (Figure 3A). The Ct value trends for ANS and NS specimens in the moderate to severely ill patients show a gradual decline from the onset of illness, with the viral load peaking around day 4 of illness. The Ct values of mildly ill patients peaked around day 3 of illness onset; moreover, they were higher on many days compared to the Ct values of moderate to severely ill patients. The Ct value of >30 of mild symptomatic patients on day 8 for both ANS and NS were earlier than those of patients with moderate to severe disease (Figure 3B). The mean Ct values of asymptomatic patients were 32.7, 32.8, and 32.6 for all specimens, ANS, and NS, respectively (Figure 3C). Asymptomatic patients were excluded from this analysis because the date of onset was unknown.

### 3.3. RSV Positivity

Of the 668 specimens, 76 ANS and 83 NS specimens tested positive for RSV by RT-PCR. The mean Ct value for RSV with positive RT-PCR results, and for ANS and NS specimens were 28.7 (range: 19.4–44.0), 28.8 (range: 19.9–40.0), and 28.6 (range: 19.4–44.0), respectively (Figure 4). The Ct values for ANS and NS specimens remained parallel over time. Comparison of the Ct values for both site specimens by day of onset shows that the viral load of ANS specimens was high on most days, except for days 0, 8, and 12 of onset and the viral levels at the ANS declined after day 10 of onset (Figure 5).

### 3.4. Evaluation of the Newly Developed Ag-RDT for SARS-CoV-2 and RSV

Table 2A shows the diagnostic performance of the RapidTesta RSV and SARS-CoV-2 test for SARS-CoV-2 using RT-PCR results as a reference. Overall, in ANS specimens, the sensitivity was 92.0% (95% CI: 86.7–95.7%) and specificity was 100% (95% CI: 97.9–100); in NS specimens, the sensitivity was 85.4% (95% CI: 79.2–90.3%) and specificity was 100% (95% CI: 97.7–100%). The PPV of RapidTesta RSV and SARS-CoV-2 compared to RT-PCR was 100%, and the NPV was 91.5–92.9%; the Cohens kappa value was 0.91–0.92. For all specimens with Ct values of <25, the sensitivity was 100% (95% CI: 94.9–100%); for Ct values 25–29, the sensitivity was 97.5% (95% CI: 92.9–99.5%); and for Ct values of ≥30, the sensitivity decreased to 70.2% (95% CI: 61.3–78.0%). The sensitivity of the test for ANS and NS samples up to 10 days after onset was 95.5% and 88.3%, respectively. The sensitivity for NS specimens within day 3 after onset was 80.4%, but this was due to the inclusion of nine early post-onset specimens with low viral loads with Ct values of ≥38 (Table 3). Specificity was 100% for both ANS and NS in all specimens (Table 2A). The RapidTesta Reader is less useful for specimens with low Ct values, that is, high viral load, in the early stages of onset; however, the test shows approximately 5% increased sensitivity for specimens with higher Ct values over time after onset. The detection limit for RapidTesta RSV and SARS-CoV-2 was a Ct value of 39.8, whereas that for RapidTesta SARS-CoV-2 was 35.3, as previously reported [11], confirming the improved detection sensitivity of RapidTesta RSV and SARS-CoV-2 for SARS-CoV-2. Thirty eight specimens tested negative by RapidTesta RSV and SARS-CoV-2 and positive by RT-PCR, that is, false negative specimens (13 ANS specimens and 25 NS specimens). The mean Ct value of the false-negative specimens was 34.5 (range: 24.4–40.5). In specimens from asymptomatic patients with unknown onset date, the mean Ct value was 32.7 (range: 24.4–40.1). The sensitivity and specificity of RapidTesta RSV and SARS-CoV-2 using RT-PCR results as a reference were 89.5% (95% CI: 66.9–98.7%) and 100% (95% CI: 15.8–100%), respectively.

The diagnostic performance of RapidTesta RSV and SARS-CoV-2 for RSV using RT-PCR results as a reference is shown in Table 2B. Overall, in ANS specimens, the sensitivity was 78.6% (95% CI: 67.1–87.5%) and specificity was 100% (95% CI: 97.4–100%); in NS specimens, the sensitivity was 77.0% (95% CI: 65.8–86.0%) and specificity was 100% (95% CI: 97.3–100%). The PPV of RapidTesta RSV and SARS-CoV-2 compared to RT-PCR was 100%, and the NPV was 77.3–88.8%; the Cohens kappa value was 0.8. The Ct values of <25, the sensitivity was 100% (95% CI: 89.1–100%); for Ct values 25–29, the sensitivity was 93.2% (95% CI: 84.7–97.7%); and for Ct values of ≥30, the sensitivity decreased to 42.6% (95% CI: 28.3–57.8%). The RapidTesta reader showed approximately 8.5% increased sensitivity for samples with Ct values > 30. Confirming the Ct values of specimens by day of onset and RapidTesta RSV and SARS-CoV-2 results, more specimens were false negative on the day of and after day 7 of onset (Figure 6). The number of false-negative specimens was 29 (14 from ANS and 15 from NS). The mean Ct value of false-negative specimens were was 34.6 (range: 27.7–44.0). From the day after to day 6 after disease onset, the sensitivity and specificity were 91.2% (95% CI: 83.4–96.1%) and 100% (95% CI: 98.1–100%), respectively, when considering the patient’ health condition.

## 4. Discussion

Studies comparing the Delta variant viral levels in the nasal cavity demonstrate that NS has high viral levels and is a useful specimen collection region for diagnosis. However, it is disadvantageous because it involves a high risk of bleeding, pain, and discomfort to the patient during specimen collection; moreover, specimen collection from NS requires expertise and involves an increased risk of infection [32]. Conversely, although ANS allows for safe specimen collection, a low viral load is a concern.

Regarding the viral dynamics of the Omicron variant, the Ct values of the ANS specimens remained close to those of the NS specimens from early in the infection to day 15 after disease onset. The Omicron variant Ct values of the ANS and NS specimens were not significantly different except on day 15 after onset. In other words, viral multiplication in the ANS was almost equivalent to that in the NS. This contrasts with the viral dynamics of the Delta variant in the ANS over time [25], suggesting a higher Omicron proliferation in the ANS. In patients with Omicron infection, self-testing results using ANS specimens are more accurate than those observed for previous prevalent variants [25]. Unlike Delta infection, Omicron infection has been shown to result in an increased dependence on cathepsin B instead of on transmembrane serine protease 2 (TMPRSS2) [33]. We speculate that it may be because of higher cathepsin B expression than TMPRSS2 expression at the ANS. A high viral load at the ANS up to 5 days after disease onset has been suggested to lead to the spread of infection [8]; however, our results, analyzed in more detail, reveal that the viral load at the ANS increases to the same level as that at the NS from early after onset. This may be involved in the spread of Omicron infection.

The WHO recommends that the criterion for the use of Ag-RDTs for suspected COVID-19 is ≤7 days after the appearance of symptoms [9]. Furthermore, because Ag-RDTs of various types and characteristics are used in many countries worldwide [8], the accuracy criteria for Ag-RDTs are that the sensitivity and specificity must be at least 80% and 97%, respectively [22]. The Ag-RDT evaluated proved to be highly accurate and useful, as sensitivity and specificity were >90% and >99%% in specimens (≤6 days after the appearance of symptoms) with various Ct values. The Cohens kappa value was 0.91–0.92, suggesting high diagnostic accuracy of the Ag-RDT for both SARS-CoV-2 antigen-positive and -negative specimens. The Ag-RDTs we evaluated in this study proved to have high diagnostic accuracy even for the Omicron variant, which carries several mutations in the viral surface spike protein structure and is currently the overwhelming mainstream of the pandemic. Furthermore, an accuracy comparison with the RapidTesta SARS-CoV-2 developed in 2021 showed that even with a Ct value of approximately 1.4 higher and a viral gene load of 4.7 × 10^0^ copies/mL lower than the limit of detection, the antigen could be accurately identified. Although the analysis of asymptomatic COVID-19 patients cannot be compared to that of symptomatic patients in terms of viral load because the time of disease onset is unknown, the Ag-RDT showed sensitivity and specificity of 89.5% and 100% for both ANS and NS, respectively. These results clearly indicate that Ag-RDT is useful for confirming viral shedding in individuals who are asymptomatic but have had close contact with COVID-19 patients; testing of such individuals would help prevent the spread of infection. This suggests that RapidTesta RSV and SARS-CoV-2 may be highly useful in evaluating for SARS-CoV-2 before or immediately after disease onset, for example, when high Ct values are expected, and when time has elapsed since the onset of the disease. Because self-testing diagnosis by Ag-RDTs using ANS specimens will become more common in the future [22], accurate diagnosis using ANS specimens with only low viral loads will continue to be in high demand. Our results demonstrate no significant differences in Omicron variant viral multiplication between ANS and NS, and validate the high diagnostic utility of ANS specimens, which can be safely self-collected. RapidTesta RSV and SARS-CoV-2 successfully identified SARS-CoV-2 antigen with high accuracy even at low viral loads, which will greatly contribute to the control of COVID-19.

RSV infects nasal epithelial cells distributed from the nasal cavity to the lower respiratory tract [34], proving that nasal aspirate is a reasonable surrogate marker for lower respiratory tract viral load [35]. However, studies comparing and evaluating RSV proliferation in ANS and NS are scarce. Our study reports, for the first time, the diagnostic utility of ANS specimens, which are safe and self-testable at the time of specimen collection, in RSV-infected patients. We identified RSV at the ANS from the day of onset, indicating that the specimens were diagnosable by Ag-RDT. The sensitivities for all specimens were lower than expected, at 78.6% and 77.0% for NS and ANS, respectively; however, they improved to 93.2% and 89.4% from day 1 through day 6 of disease onset. RSV load in the nasal cavity was low on the day of and after day 7 of onset, and Ag-RDT testing should be fully considered during this period of low viral load. In other words, using an Ag-RDT on ANS samples to diagnose RSV infection is useful and effective in routine outpatient care. However, false-negative results must also be considered when patients present to the emergency department within a few hours after the onset or when RSV infection is ruled out by respiratory symptoms or fever that persists for about a week.

The RapidTesta RSV and SARS-CoV-2 analyzed in this study successfully identified two pathogens, RSV and SARS-CoV-2. To maximize the usefulness of this Ag-RDT, the following virological dynamics must be recognized: COVID-19 can be accurately diagnosed on the first day of illness, even several days after disease onset. In contrast, for RSV infection, caution should be exercised in testing on the days 0 and 7 after disease onset, when the viral load is low. Furthermore, detection of RSV in ANS specimens clearly indicates that it is highly useful for patients with suspected COVID-19 or RSV infection to self-test. RSV and SARS-CoV-2 were detected simultaneously in five patients; four patients were under 5 years old and one patient was 71 years old. Due to the small number of cases, we were unable to evaluate basic analyses, such as the interaction of viral growth in the nasal cavity and comparison of clinical severity.

This study had two limitations. As the SARS-CoV-2 mutant variant being evaluated was the Omicron variant, the virology of other mutant variants and the accuracy of RapidTesta RSV and SARS-CoV-2 are unclear. Second, we have not validated the Ag-RDT using samples with conditions or interactions that could lead to the false positive or the false negative, such as blood or bacterial contaminated samples.

## 5. Conclusions

In summary, this study evaluated the Omicron variant of SARS-CoV-2 and RSV in nasal specimens. The viral load of SARS-CoV-2 tended to be higher in the NS than in the ANS over time after onset, although the difference in viral load between the two sites was small. Thus, COVID-19 diagnosis by Ag-RDT using ANS specimens was successfully validated. For RSV infection, analysis of the viral load in ANS and NS specimens over time revealed that despite lower viral loads in the ANS than in the NS, continuous excretion of virus was observed at the ANS. Although caution should be exercised in diagnosing Ag-RDT on the day of onset when the viral load is low or after some time has elapsed since onset, we consider Ag-RDT to be highly useful in general practice.

## Figures and Tables

**Figure 1 diagnostics-14-00119-f001:**
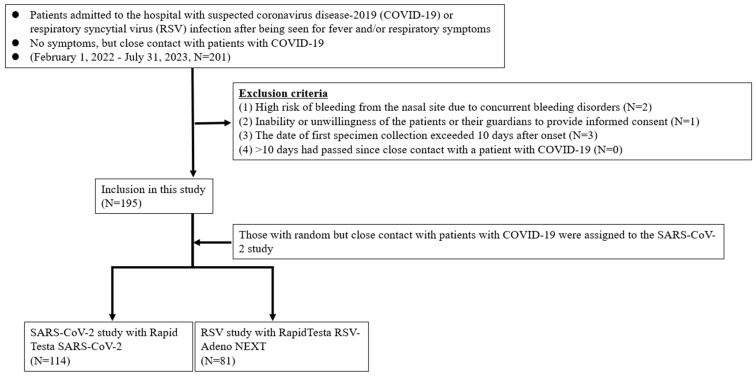
Flowchart for the selection of the study population.

**Figure 2 diagnostics-14-00119-f002:**
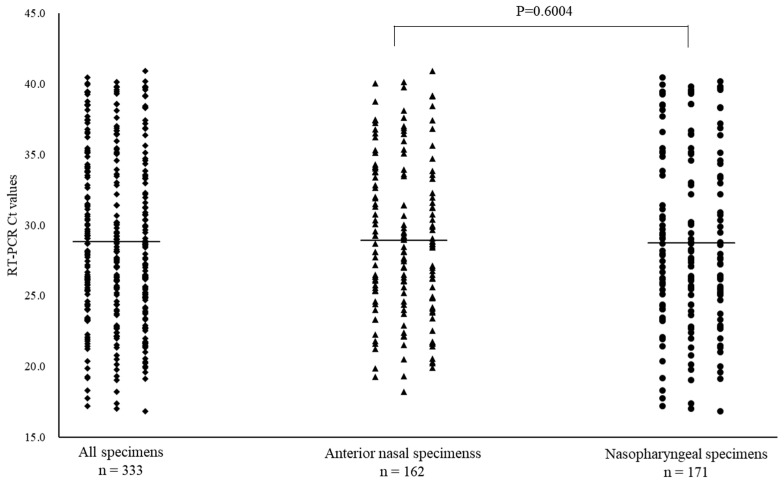
Comparison of cycle threshold (Ct) values for all specimens, anterior nasal, and nasopharyngeal specimens for coronavirus disease-2019. Mean Ct values are indicated by horizontal bars.

**Figure 3 diagnostics-14-00119-f003:**
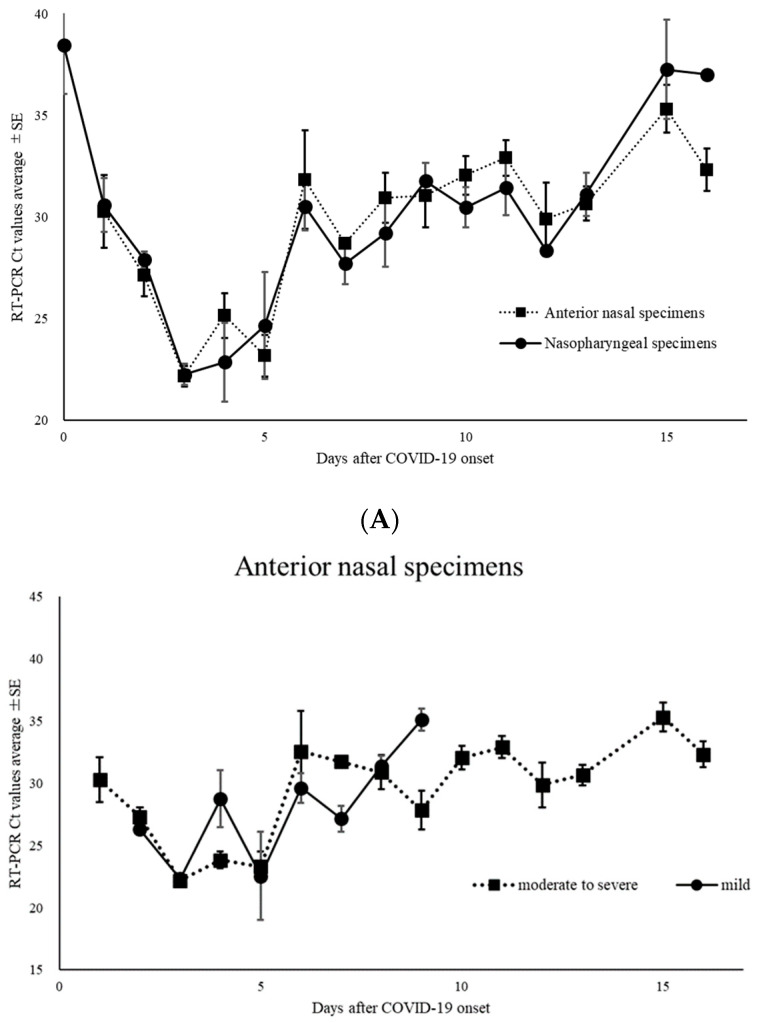
(**A**) Changes over time in thresholds (Ct) of nasopharyngeal and anterior nasal specimens in all coronavirus disease-2019; (**B**) changes over time in the cycle threshold (Ct) of nasopharyngeal and anterior nasal specimens in patients with mild and moderate to severe coronavirus disease-2019; (**C**) comparison of cycle threshold (Ct) values for all specimens, anterior nasal, and nasopharyngeal specimens taken from asymptomatic patients for coronavirus disease-2019. Mean Ct values are indicated by horizontal bars.

**Figure 4 diagnostics-14-00119-f004:**
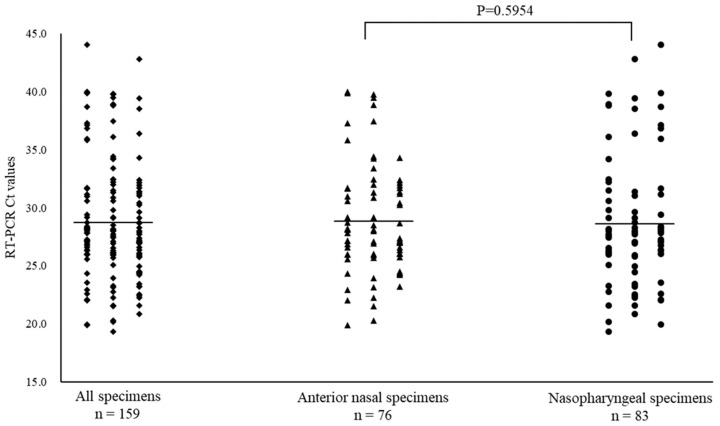
Comparison of cycle threshold (Ct) values for all specimens, anterior nasal, and nasopharyngeal specimens for the respiratory syncytial virus. Mean Ct values are indicated by horizontal bars.

**Figure 5 diagnostics-14-00119-f005:**
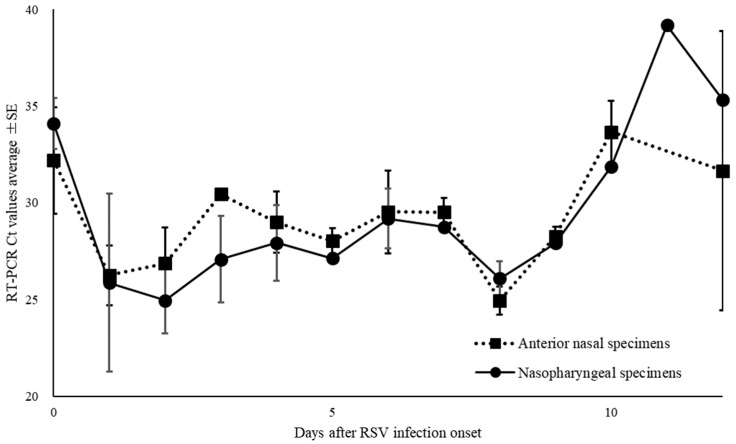
Changes over time in thresholds (Ct) of nasopharyngeal and anterior nasal specimens in respiratory syncytial virus.

**Figure 6 diagnostics-14-00119-f006:**
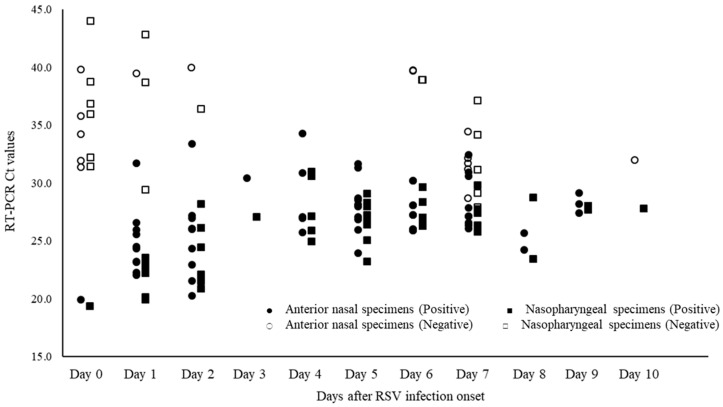
Ct values of respiratory syncytial virus (RSV) specimens by date of onset and determination of RapidTesta RSV and SARS-CoV-2.

**Table 1 diagnostics-14-00119-t001:** Distribution of severity of coronavirus disease-2019 (COVID-19) and respiratory syncytial virus (RSV) patients.

Clinical Severity	COVID–19 (*n* = 56)	RSV (*n* = 47)
Moderate to Severe Age average (y) Age range (y)	10 21.9 0–79	13 3.4 0–15
Mild Age average (y) Age range (y)	28 31.3 2–71	34 26.4 0–71
Asymptomatic Age average (y) Age range (y)	18 28.9 6–72	– – –

**Table 2 diagnostics-14-00119-t002:** (**A**) Accuracy of SARS-CoV-2 diagnosis in the two-antigen-detecting rapid diagnostic test; (**B**) accuracy of respiratory syncytial virus (RSV) diagnosis in the two-antigen-detecting rapid diagnostic test.

(A)						
		Sensitivity (%)	95%CI	Specificity (%)	95%CI	Detection limit by Ct value
RapidTesta RSV & SARS-CoV-2 (visual judgement)	Anterior nasal specimens	92.0% (149/162)	86.7–95.7%	100% (171/171)	97.9–100%	39.8
	Nasopharyngeal specimens	85.4% (146/171)	79.2–90.3%	100% (162/162)	97.7–100%	39.9
RapidTesta RSV & SARS-CoV-2 (RapidTesta Reader)	Anterior nasal specimens	92.6% (150/162)	87.4–96.1%	99.4% (169/170)	96.8–100%	39.8
	Nasopharyngeal specimens	86.5% (148/171)	80.5–91.3%	100% (161/161)	97.7–100%	40.2
RapidTesta SARS-CoV-2 (visual judgement)	Anterior nasal specimens	92.1% (140/152)	86.6–95.9%	100% (66/66)	94.6–100%	38.4
	Nasopharyngeal specimens	86.0% (141/164)	79.7–90.9%	100% (59/59)	93.9–100%	39.3
RapidTesta SARS-CoV-2 (RapidTesta Reader)	Anterior nasal specimens	92.2% (130/141)	86.5–96.0%	100% (65/65)	94.5–100%	37.6
	Nasopharyngeal specimens	88.5% (131/148)	82.2–93.2%	100% (58/58)	93.8–100%	38.8
**(B)**						
		**Sensitivity (%)**	**95%CI**	**Specificity (%)**	**95%CI**	**Detection limit by Ct value**
RapidTesta RSV & SARS-CoV-2 (visual judgement)	Anterior nasal specimens	78.6% (55/70)	67.1–87.5%	100% (139/139)	97.4–100%	34.3
	Nasopharyngeal specimens	77.0% (57/74)	65.8–86.0%	100% (135/135)	97.3–100%	31.0
RapidTesta RSV & SARS-CoV-2 (RapidTesta Reader)	Anterior nasal specimens	80.0% (56/70)	68.7–88.6%	100% (139/139)	97.4–100%	34.3
	Nasopharyngeal specimens	79.7% (59/74)	68.8–88.2%	99.3% (134/135)	95.9–100%	34.2
RapidTesta SARS-CoV-2 (visual judgement)	Anterior nasal specimens	74.6% (50/67)	62.5–84.5%	100% (43/43)	91.8–100%	34.3
	Nasopharyngeal specimens	74.6% (53/71)	62.9–84.2%	100% (39/39)	91.0–100%	31.0
RapidTesta SARS-CoV-2 (RapidTesta Reader)	Anterior nasal specimens	80.6% (54/67)	69.1–89.2%	100% (43/43)	91.8–100%	34.3
	Nasopharyngeal specimens	77.5% (55/71)	66.0–86.5%	100% (39/39)	91.0–100%	31.0

**Table 3 diagnostics-14-00119-t003:** Overview of the sensitivity of the antigen-detecting rapid diagnostic test for SARS-CoV-2 by the post-onset course of coronavirus disease (COVID-19).

		≤3 days		4–6 days		7–10 days		≥11 days	
		Sensitivity (%)	95%CI	Sensitivity (%)	95%CI	Sensitivity (%)	95%CI	Sensitivity (%)	95%CI
RapidTesta RSV & SARS-CoV-2 (visual judgement)	Anterior nasal specimens	97.6% (41/42)	(87.4–99.9%)	93.3% (28/30)	(77.9–99.2%)	94.9% (37/39)	(82.7–99.4%)	80.0% (24/30)	(61.4–92.3%)
	Nasopharyngeal specimens	80.4% (41/51)	(66.9–90.2%)	90.3% (28/31)	(74.2–98.0%)	97.4% (37/38)	(86.2–99.9%)	70.0% (21/30)	(50.6–85.3%)
RapidTesta RSV & SARS-CoV-2 (RapidTesta Reader)	Anterior nasal specimens	97.6% (41/42)	(87.4–99.9%)	93.3% (28/30)	(77.9–99.2%)	94.9% (37/39)	(82.7–99.9%)	83.3% (25/30)	(65.3–94.4%)
	Nasopharyngeal specimens	80.4% (41/51)	(66.9–90.2%)	90.3% (28/31)	(74.2–98.0%)	97.4% (37/38)	(86.2–99.9%)	76.3% (23/30)	(57.7–90.1%)
RapidTesta SARS-CoV-2 (visual judgement)	Anterior nasal specimens	95.1% (39/41)	(83.5–99.9%)	100% (24/24)	(85.8–100%)	91.4% (32/35)	(76.9–98.2%)	73.3% (22/30)	(54.1–87.7%)
	Nasopharyngeal specimens	85.1% (40/47)	(71.1–93.8%)	96.3% (26/27)	(81.8–99.9%)	91.7% (33/36)	(77.5–98.2%)	66.7% (20/30)	(47.2–82.7%)
RapidTesta SARS-CoV-2 (RapidTesta Reader)	Anterior nasal specimens	95.1% (39/41)	(88.4–100%)	100% (24/24)	(85.8–100%)	92.6% (25/27)	(75.5–99.1%)	77.8% (21/27)	(57.7–91.4%)
	Nasopharyngeal specimens	87.2% (41/47)	(74.3–95.2%)	96.2% (25/26)	(86.8–100%)	93.3% (28/30)	(77.9–99.2%)	66.7% (18/27)	(46.0–83.5%)

## Data Availability

The data presented in this study are available on request from the corresponding author. The data are not publicly available due to arrangements made by the Ethics Committee.

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
