# Peer review of "Temporal Trend of the SARS-CoV-2 Omicron Variant and RSV in the Nasal Cavity and Accuracy of the Newly Developed Antigen-Detecting Rapid Diagnostic Test"

_diagnostics, 2024, doi:10.3390/diagnostics14010119_

Round 1

Reviewer 1 Report (New Reviewer)

Comments and Suggestions for Authors

The title is appropriate for the content of the article.

The summary is well-written and includes all the main results. I suggest to the authors:

      Instead of the full name of respiratory syncytial virus in the title, I suggest replacing it with the abbreviation RSV (which makes the title more compact).

The introduction is appropriate for the results presented below.

I recommend the authors:

To include the role of RSV with co-infections with SARS-CoV-2 in different age groups.

1. In which age groups is there the highest rate of mixed infections, will the results of this article contribute to solving the problem of diagnosis of mixed viral infections?

There is a lack of precisely stated originality and contributions in the exposition.

The figures and tables reflect the results achieved well.

In the conclusion, the achieved results of the set goals are summarized.

Author Response

To Reviewer 1,

Thank you for taking the time to review our manuscript and for your remarks. We will answer the questions from Reviewer 1 accurately and clearly. Changes made in the manuscript are reflected in the text as yellow highlighting.

The title is appropriate for the content of the article.

The summary is well-written and includes all the main results. I suggest to the authors: Instead of the full name of respiratory syncytial virus in the title, I suggest replacing it with the abbreviation RSV (which makes the title more compact).

Response: As reviewer 1 indicated, we have changed the title of the manuscript with RSV as follows “Temporal trend of the SARS-CoV-2 Omicron variant and RSV in the nasal cavity and accuracy of the newly developed antigen-detecting rapid diagnostic test”

The introduction is appropriate for the results presented below.

I recommend the authors: To include the role of RSV with co-infections with SARS-CoV-2 in different age groups. 1. In which age groups is there the highest rate of mixed infections, will the results of this article contribute to solving the problem of diagnosis of mixed viral infections? There is a lack of precisely stated originality and contributions in the exposition.

Response: Thank you for pointing this out.There were a total of 5 patients with mixed RSV and SARS-CoV-2 infections: four patients were under 5 years old and 1 patient was 71 years old. Unfortunately, the small number of patients with mixed infections precludes statistical analysis. It is difficult to conclude from the results of this study that mixed SARS-CoV-2 and RSV infections may tend to occur in children, who are more prone to RSV infections. The following commentary is added to the manuscript in Line 524: RSV and SARS-CoV-2 were detected simultaneously in 5 patients; 4 patients were under 5 years old and 1 patient was 71 years old. Due to the small number of cases, we were unable to evaluate basic analyses, such as the interaction of viral growth in the nasal cavity and comparison of clinical severity.

The figures and tables reflect the results achieved well. In the conclusion, the achieved results of the set goals are summarized.

Response: Thank you for your comments on Figure and the summary in the manuscript.

Reviewer 2 Report (New Reviewer)

Comments and Suggestions for Authors

I read this article with great interest that the authors aimed to evaluate and validate the clinical performance of the newly developed rapid antigen test for SARS-CoV-2 and RSV (Ag-RDT). The authors collected both ANS and NS specimens for Ag-RDT and found that there were good correlations between the results of viral load and antigen expression. These results also promised the domestic use of rapid antigen test by people who are not healthcare providers. This manuscript is well-written, and I have only few comments.

1. In Figure 1, the arrow linked with the section of exclusion criteria should be right arrow, not left arrow.

2. The authors mentioned that 114 patients participated in the SARS-CoV-2 group and 81 patients participated in the RSV group in the Results section (line 235-236). However, it was shown that 112 patients participated in the SARS-CoV-2 group and 83 patients participated in the RSV group in Figure 1. Please make sure.

3. The authors described that 4 adult patients with COVID-19 and 19 pediatric patients (6 with COVID-19 and 13 with RSV infection) were hospitalized for moderate to severe disease (line 240-241). Therefore, the case number of moderate to severe COVID-19 should be 10. However, there were 15 cases with moderate to severe COVID-19 shown in Table 1. Please make sure.

4. Did the authors validate any potential conditions or interactions that could lead to falsely positive or falsely negative results in Ag-RDT (For example, bloody specimens, other bacterial, viral or fungal antigens in the specimen)? If not, consider discussing these as the limitations of this study in the Discussion section. 

Author Response

To Reviewer 2,

Thank you for taking the time to review our manuscript and for your remarks. We will answer the questions from Reviewer 2 accurately and clearly. Changes made in the manuscript are reflected in the text as yellow highlighting.

I read this article with great interest that the authors aimed to evaluate and validate the clinical performance of the newly developed rapid antigen test for SARS-CoV-2 and RSV (Ag-RDT). The authors collected both ANS and NS specimens for Ag-RDT and found that there were good correlations between the results of viral load and antigen expression. These results also promised the domestic use of rapid antigen test by people who are not healthcare providers. This manuscript is well-written, and I have only few comments.

  1. In Figure 1, the arrow linked with the section of exclusion criteria should be right arrow, not left arrow.

Response: Thank you for pointing this out. We have changed to a right-pointing arrow.

  1. The authors mentioned that 114 patients participated in the SARS-CoV-2 group and 81 patients participated in the RSV group in the Results section (line 235-236). However, it was shown that 112 patients participated in the SARS-CoV-2 group and 83 patients participated in the RSV group in Figure 1. Please make sure.

Response: We are terribly sorry. The numbers in the manuscript were correct and the numbers in the Figure 1 were incorrect. Therefore, we have changed the numbers in the Figure 1 to the correct ones.

  1. The authors described that 4 adult patients with COVID-19 and 19 pediatric patients (6 with COVID-19 and 13 with RSV infection) were hospitalized for moderate to severe disease (line 240-241). Therefore, the case number of moderate to severe COVID-19 should be 10. However, there were 15 cases with moderate to severe COVID-19 shown in Table 1. Please make sure.

Response: We are terribly sorry. The numbers in the manuscript were correct and the numbers in the Table 1 were incorrect. Therefore, we have changed the numbers in the Table 1 to the correct ones.

  1. Did the authors validate any potential conditions or interactions that could lead to falsely positive or falsely negative results in Ag-RDT (For example, bloody specimens, other bacterial, viral or fungal antigens in the specimen)? If not, consider discussing these as the limitations of this study in the Discussion section. 

Response: Thank you for pointing this out. In the Ag-RDT we evaluated in this study, we did not examine conditions or interactions that could result in the false positive or the false negative. Therefore, we have added your suggestion as a limitation.

This manuscript is a resubmission of an earlier submission. The following is a list of the peer review reports and author responses from that submission.

Round 1

Reviewer 1 Report

Comments and Suggestions for Authors

In the manuscript by Tamura et al., the authors analyzed the viral load and elimination period of SARS-CoV- 2 in the nasal cavity during the Omicron epidemic over time and evaluated a newly developed Ag-RDT that can simultaneously and rapidly detect the presence of SARS-CoV-2 and RSV antigens in a single specimen.

Overall, the manuscript showed several critical limitations, especially concerning the validation of the combined SARS-CoV-2 and RSV Ag-RDT.

Major:

  1. It seems that the validation of individual tests has already been carried out as reported in the materials and methods (refs. 11-12, “The evaluation data for the 117 respective Ag-RDTs has shown that RapidTesta SARS-CoV-2 has a sensitivity of 89.5% 118 and specificity of 100% [11], whereas RapidTesta RSV-Adeno NEXT shows 98.1% and 119 100% for RSV antigen detection, respectively [12]). Is a new validation required?

  2. The number of patients is very limited for both the aims of the paper.

  3. The sampling made by the same physician doesn’t reflect the real-world practice and the aim of the

    Ag-RDT (self-collection and analysis from the patients).

  4. The author must include the sensitivity and specificity of the RT-PCR kits used for this study.

  5. What are the technical differences between RapidTesta

    RSV & SARS-CoV-2 and RapidTesta SARS-CoV-2 in the detection of SARS-CoV-2? It seems that there

    are differences in the sensitivity.

  6. The data analysis was minimal; no PPV, NPV, or Cohens kappa values were not mentioned. These

    detailed statistical data should be a must for this type of article.

  7. As stated in the materials and methods section (“All patients had upper respiratory tract inflammatory

    symptoms such as sore throat 178 and dry cough on admission, and 50% had lower respiratory tract inflammatory 179 findings, as confirmed by computed tomography images. All adult patients were on 180 ventilator management because of deteriorating respiratory status after admission and 181 were considered severe. Among the six pediatric COVID-19 patients, two were 182 considered severe, one with encephalopathy, and the other with pneumonia requiring 183 oxygen supplementation. Other pediatric patients were admiged in moderate condition 184 for symptoms such as febrile convulsions, croup syndrome, and moderate dehydration.”), the study did not include samples from asymptomatic and paucisymptomatic patients, but only from symptomatic patients, including patients with severe symptoms. This aspect should be understandable for the evaluation of viral load and elimination period SARS-CoV-2 in the nasal cavity, but not for the validation of the novel Ag- RDT (bias). These two groups (asymptomatic and paucisymptomatic) represent the challenging issues of Ag-RDT. Indeed, as reported in the manuscript, the reduction of the viral load in the patients analyzed reduced the sensitivity of the test (Table 2).

  8. The collection, although carried out on different days, and the analysis of multiple samples from the same positive patients represent another bias for the statistical analysis.

  9. Similarly, the mean value of Ct is altered by the analysis of samples from patients after several days of symptoms. The reduction of the viral load clearly impacts and alters the Ct.

Comments on the Quality of English Language

Minor editing of English language required.

Author Response

To Reviewer 1,

Major:

  1. It seems that the validation of individual tests has already been carried out as reported in the materials and methods (refs. 11-12, “The evaluation data for the 117 respective Ag-RDTs has shown that RapidTesta SARS-CoV-2 has a sensitivity of 89.5% 118 and specificity of 100% [11], whereas RapidTesta RSV-Adeno NEXT shows 98.1% and 119 100% for RSV antigen detection, respectively [12]). Is a new validation required?

Response: Thank you for taking the time to review our manuscript and for your remarks. RapidTesta SARS-CoV-2 is an Ag-RDT that detects SARS-CoV-2 antigen and RapidTesta RSV-Adeno NEXT is an Ag-RDT that detects RSV antigen. Both have been approved by the Ministry of Health, Labor and Welfare and are already in clinical use. The RapidTesta RSV & SARS-CoV-2 evaluated in this study is based on the validated RapidTesta SARS-CoV-2 and RapidTesta RSV-Adeno NEXT technologies. In combining these two Ag-RDTs, a reevaluation of reagent performance is necessary since many improvements (e.g., adjustment of the buffer for suspending swab specimens, optimization of the membrane for visualizing the antigen-antibody reaction, etc.) have been made based on the latest findings. Furthermore, combining two different Ag-RDTs that detect a single antigen each into a single Ag-RDT that detects both antigens may lead to unanticipated nonspecific reactions. Therefore, it is actually necessary to evaluate the accuracy of the improved Ag-RDT using new clinical specimens. For this reason, we conducted the present study.

  1. The number of patients is very limited for both the aims of the paper.

Response: Thank you for pointing this out. The COVID-19 patients who participated in this study were only moderately to severely ill patients; asymptomatic and mildly ill patients were unavailable, and hence, not included. Therefore, the results of the analysis of nasal virus dynamics revealed in this study focus on the transition of patients with moderate to severe disease and do not fully reflect that of asymptomatic and mildly ill patients. Furthermore, the number of SARS-CoV-2 antigen-negative specimens was low. Fewer antigen-negative specimens may lead to an underestimation of the nonspecific response of Ag-RDT and, consequently, to a higher specificity. In summary, a limitation of this study is the clinical predominance of patients with moderate to severe disease, the lack of asymptomatic or mildly symptomatic patients, and the small number of antigen-negative patients. To clarify the same for benefit of the reader, we have added the following text to the limitation section of the text.

Discussion, Lines 467–470

This study has three limitations. The participants in this study were mainly adults and children with moderate to severe disease, and were few in number. Therefore, our analysis of nasal viral dynamics does not reflect the viral dynamics of all Omicron variant-infected patients, including asymptomatic and mildly ill patients.

In addition, we made the following changes to the text describing the nasal viral dynamics in the Results and Discussion sections to avoid any misunderstanding by the reader.

Results, Lines 248–250

Of all 228 specimens collected mainly from patients with moderate to severe disease, the number of specimens positive for SARS-CoV-2 by RT-PCR was 111 for anterior nasal specimens and 111 for nasopharyngeal specimens.

Discussion, Lines 407–409

In the viral dynamics of the Omicron variant, mainly in patients with moderate to severe disease, the Ct values of the anterior nasal specimens remained close to those of the nasopharyngeal specimens from early onset to day 15 of disease onset.

On the other hand, the specimens we analyzed were always collected from the nasal vestibular and nasopharyngeal sites at the same time, for a total of 228 specimens. Furthermore, since we collected specimens from the anterior nasal and nasopharyngeal sites at different times after disease onset, we were able to collect specimens with various virus levels. Therefore, although the small number of patients is a limitation, we were able to analyze a large number of specimens containing virus amounts just below the detection limit for Ag-RDT evaluation, and we believe that we were able to accurately evaluate the sensitivity of the Ag-RDT.

3.The sampling made by the same physician doesn’t reflect the real-world practice and the aim of the Ag-RDT (self-collection and analysis from the patients).

Response: Thank you for your comments and suggestions.

The WHO has shown the usefulness of self-testing in COVID-19 diagnosis using Ag-RDT. However, it has not been clear how the Omicron viral load changes over time in the anterior nasal cavity, which is one of the sampling sites used for self-testing. Therefore, to address the question of whether the viral load at the anterior nasal site is appropriate as a specimen collection site, we first conducted this study to evaluate the exact Omicron viral dynamics in the anterior nasal cavity nasal site and nasopharyngeal site.

As you commented, self-testing results can vary depending on how specimens are collected by individuals. Of course, even medical professionals will have varied methods of specimen collection. For this reason, the instructions for use of the newly developed Ag-RDT analyzed in this study illustrate how to collect specimens by self-testing from the anterior nasal site, referring to the guidelines of the National Institute of Infectious Diseases (ref 1). Through our manuscript, we hope that the usefulness of collecting specimens from the anterior nasal cavity, as recommended by the National Institute of Infectious Diseases and WHO (ref 2), will be widely recognized, accurate specimen collection methods will be understood, and the results will be useful for self-testing in COVID-19 diagnosis.

Ref 1. National Institute of Infectious Diseases, Coronavirus infections disease (COVID-19) Guideline for pathogen testing. https://www.mhlw.go.jp/content/001029252.pdf

Ref 2. WHO, Use of SARS-CoV-2 antigen-detection rapid diagnostic tests for COVID-19 self-testing.

https://www.who.int/publications/i/item/WHO-2019-nCoV-Ag-RDTs-Self_testing-2022.1

  1. The author must include the sensitivity and specificity of the RT-PCR kits used for this study.

Response: Thank you for your comments. We followed the guidelines (ref 3) developed by the National Institute of Infectious Diseases for the viral RNA extraction kits used for the RT-PCR method in this study, as well as for the enzymes, primers, and programs used in the RT-PCR method and their use. The guidelines of the National Institute of Infectious Diseases comply with the WHO standards for the use of RT-PCR methods. Therefore, we did not seek to determine the sensitivity and specificity of the RT-PCR method itself, but used the results of the RT-PCR method as the gold standard to compare the results of the newly developed Ag-RDT.

Ref 3. National Institute of Infectious Diseases, Coronavirus disease (COVID-19)

https://www.niid.go.jp/niid/ja/lab-manual-m/9559-2020-04-14-10-09-54.html

  1. What are the technical differences between RapidTesta RSV & SARS-CoV-2 and RapidTesta SARS-CoV-2 in the detection of SARS-CoV-2? It seems that there are differences in the sensitivity.

Response: Thank you for your question. RapidTesta SARS-CoV-2 and RapidTesta RSV & SARS-CoV-2 are both Ag-RDTs that detect SARS-CoV-2 antigen. After the commercialization of RapidTesta SARS-CoV-2, we made technological innovations to improve the detection accuracy, such as the solution to dilute the swab specimen, the filter used to drop the specimen solution, and the membrane to make the antigen-antibody reaction visible. Since we are developing this technology as a product patent, it is difficult to go into details, but we have made various improvements to increase the detection accuracy.

  1. The data analysis was minimal; no PPV, NPV, or Cohens kappa values were not mentioned. These detailed statistical data should be a must for this type of article.

Response: Thank you for your comment. In our study, the positive predictive value (PPV) of RapidTesta RSV & SARS-CoV-2 (visual judgement) compared to RT-PCR was 100%, and the negative predictive value (NPV) was 33.3%. The Cohens kappa value was 0.48. These results were added in the results section as follows.

Results, Lines 363-365

The positive predictive value (PPV) of RapidTesta RSV & SARS-CoV-2 (visual judgement) compared to RT-PCR was 100%, and the negative predictive value (NPV) was 33.3%; the Cohens kappa value was 0.48.

The Cohens kappa value was 0.48, which could be attributed to the low number of negative specimens, although a sufficient number of SARS-CoV-2 positive specimens could be evaluated.

Discussion, Lines 448-450

The Cohens kappa value was 0.48, which could be attributed to the low number of negative specimens, although a sufficient number of SARS-CoV-2 positive specimens was evaluated.

  1. As stated in the materials and methods section (“All patients had upper respiratory tract inflammatory symptoms such as sore throat 178 and dry cough on admission, and 50% had lower respiratory tract inflammatory 179 findings, as confirmed by computed tomography images. All adult patients were on 180 ventilator management because of deteriorating respiratory status after admission and 181 were considered severe. Among the six pediatric COVID-19 patients, two were 182 considered severe, one with encephalopathy, and the other with pneumonia requiring 183 oxygen supplementation. Other pediatric patients were admiged in moderate condition 184 for symptoms such as febrile convulsions, croup syndrome, and moderate dehydration.”), the study did not include samples from asymptomatic and paucisymptomatic patients, but only from symptomatic patients, including patients with severe symptoms. This aspect should be understandable for the evaluation of viral load and elimination period SARS-CoV-2 in the nasal cavity, but not for the validation of the novel Ag- RDT (bias). These two groups (asymptomatic and paucisymptomatic) represent the challenging issues of Ag-RDT. Indeed, as reported in the manuscript, the reduction of the viral load in the patients analyzed reduced the sensitivity of the test (Table 2).

Response: Thank you for your comments. The patients included in this study were mainly patients with moderate or severe disease and few patients with asymptomatic or mild disease. Therefore, the nasal viral dynamics of the Omicron variant revealed in our study were the results of an analysis focusing on COVID-19 patients with moderate or severe disease, and not the viral dynamics that also included asymptomatic or mildly disease patients. We have added a note on lines 248-250 in the Results section, and lines 467-470 in the Discussion section for increased clarity for the reader.

As you commented, the viral load varies depending on disease severity (ref 4). An important factor involved in the diagnostic accuracy of Ag-RDT is the viral load. In other words, a higher viral load increases sensitivity and vice-versa. Therefore, symptoms and disease severity do not directly affect the diagnostic accuracy of Ag-RDT. In our study, we collected specimens with various virus amounts to evaluate the accuracy of the Ag-RDT diagnostic system; since some specimens with high Ct values, closer to the detection limit of Ag-RDT, were included, we believe that we were able to accurately evaluate the accuracy of Ag-RDT.

Ref 4. Dadras O. The relationship between COVID-19 viral load and disease severity: A systematic review. Immun Inflamm Dis. 2022

  1. The collection, although carried out on different days, and the analysis of multiple samples from the same positive patients represent another bias for the statistical analysis.

Response: Thank you very much for your insightful remark. As you stated, if the number of specimens collected differs greatly from day to day, even if specimens are collected on different days after disease onset, an average value may also cause a bias in the statistical analysis. Therefore, as shown in Table 2, we suppressed the effects of intra-day variation by dividing the specimen collection days into 3-day increments and tabulating the number of specimens collected on each day. Because the number of specimens in each group ranged from 25 to 30, we believe that there is no significant variation.

  1. Similarly, the mean value of Ct is altered by the analysis of samples from patients after several days of symptoms. The reduction of the viral load clearly impacts and alters the Ct.

Response: Thank you for your remark.

The viral load and Ct values are proven to correspond (ref 5). Our study results showed that nasal Ct levels peaked on days 3 to 5 of illness onset and then gradually increased. This study suggests that the timing of peak Ct values in patients with moderate to severe disease after Omicron variant infection may be earlier than that for the Delta variant.

Ref 5. Aoki K et al. Clinical validation of quantitative SARS-CoV-2 antigen assays to estimate SARS-CoV-2 viral loads in nasopharyngeal swabs. J Infect Chemother. 2021.

Reviewer 2 Report

Comments and Suggestions for Authors

After almost 3 year of pandemia and one year later of the new normality, from my point of view the paper is absoleted. 

- It is nothing new neither novelty

-nowdays in Hospiatal we have a lot of IC rapid test for repsiratory virus besides the rt/PCR

- The major problem is the N, only 10 patients, when usual the minimun N is 100, -for this kind of stiudy and under emergency circumstances as the pandemia-, ohterwise the N is > 100.

Only this invalidate the statistical analysis as well as the conclusions.

Author Response

To Reviewers 2,

- It is nothing new neither novelty

-now days in Hospital we have a lot of IC rapid test for respiratory virus besides the rt/PCR

- The major problem is the N, only 10 patients, when usual the minimum N is 100,

-for this kind of study and under emergency circumstances as the pandemic-, otherwise the N is > 100.

Only this invalidate the statistical analysis as well as the conclusions.

- It is nothing new neither novelty

-now days in Hospital we have a lot of IC rapid test for respiratory virus besides the rt/PCR

Response: Thank you for taking the time to review our manuscript and your comments.

As you stated, many Ag-RDTs are currently used in clinical practice to identify various pathogen antigens. SARS-CoV-2 mutates rapidly, and in the 3 years since the pandemic, various variants have occurred, from Alpha to Omicron variant. It is unclear whether the structural variants of the viral surface protein and the viral gene mutations encoding these variants, which are often present in Omicron, have an impact on the diagnostic efficacy of Ag-RDT. The World Health Organization states that the accuracy of the newly developed Ag-RDT should be evaluated using a clinical specimen (ref 6). Furthermore, the Ag-RDT that was evaluated in our study is a technical modification of an existing Ag-RDT that detects SARS-CoV-2 antigen. In addition, we made technical improvements to an existing Ag-RDT for detecting RSV antigen and combined these two antigen detection technologies into a single Ag-RDT. As a result, we were able to demonstrate that the newly developed Ag-RDT is capable of diagnosing Omicron (B.1.1.529) with high accuracy.

Ref 6. WHO, Use of SARS-CoV-2 antigen-detection rapid diagnostic tests for COVID-19 self-testing. Interim guidance 9 March 2022

https://www.who.int/publications/i/item/WHO-2019-nCoV-Ag-RDTs-Self_testing-2022.1

- The major problem is the N, only 10 patients, when usual the minimum N is 100,

-for this kind of study and under emergency circumstances as the pandemic-, otherwise the N is > 100.

Response: Thank you for pointing this out. The patients with COVID-19 who participated in this study were only moderately to severely ill; asymptomatic and mildly ill patients were unavailable, and hence, not included in the study. Therefore, the results of the analysis of nasal virus dynamics in this study focused on the transition of patients with moderate to severe disease and do not fully reflect that of asymptomatic and mildly ill patients. Furthermore, the number of SARS-CoV-2 antigen-negative specimens was low. Fewer antigen-negative specimens may lead to an underestimation of the nonspecific response of the Ag-RDT and, consequently, to a higher specificity. In summary, a limitation of this study is the clinical predominance of patients with moderate to severe disease, the lack of asymptomatic or mildly symptomatic patients, and the small number of antigen-negative patients. To improve clarity, we have added the following text to the limitation section of the text.

Discussion, Lines 467–470

This study has three limitations. The participants in this study were mainly adults and children with moderate to severe disease, and were few in number. Therefore, our analysis of nasal viral dynamics does not reflect the viral dynamics of all Omicron variant-infected patients, including asymptomatic and mildly ill patients.

In addition, we made the following changes to the text describing the nasal viral dynamics in the Results and Discussion sections to avoid any misunderstanding by the reader.

Results, Lines 248–250

Of all 228 specimens collected mainly from patients with moderate to severe disease, the number of specimens positive for SARS-CoV-2 by RT-PCR was 111 for anterior nasal specimens and 111 for nasopharyngeal specimens.

Discussion, Lines 407–409

In the viral dynamics of the Omicron variant, mainly in patients with moderate to severe disease, the Ct values of the anterior nasal specimens remained close to those of the nasopharyngeal specimens from early onset to day 15 of disease onset.

On the other hand, the specimens we analyzed were always collected from the nasal vestibular and nasopharyngeal sites at the same time, for a total of 228 specimens. Furthermore, since we collected specimens from the anterior nasal and nasopharyngeal sites at different times after disease onset, we were able to collect specimens with various virus levels. Therefore, although the small number of patients is a limitation, we were able to analyze a large number of specimens containing virus amounts just below the detection limit for Ag-RDT evaluation, and we believe that we were able to accurately evaluate the sensitivity of the Ag-RDT.

Reviewer 3 Report

Comments and Suggestions for Authors

The manuscript that I reviewed “Temporal trend of SARS-CoV-2 Omicron variant in the nasal cavity and accuracy of the newly developed antigen-detecting rapid diagnostic test” is a study aimed to evaluate the accuracy of an Ag-RDT that can concurrently detect SARS-CoV-2 and respiratory syncytial virus (RSV) antigens in nasal specimens previously tested by real time RT-PCR and for which was hence known viral load and elimination period. The Authors conclude that the Ag-RDT had stable sensitivity and high specificity even at high Ct values and anterior nasal specimens resulted to be more useful in the Omicron variant epidemic, also being more accurate than the existing rapid  diagnosis kits. Furthermore, the anterior nasal specimens resulted more useful than nasopharyngeal at least for the Omicron variant epidemic.

Major comments

The article is well written and interesting. The statements defined are supported by detailed presented data. In my opinion, a weakness of this work is represented by the description of sampling that reports detail about the disease but it is not clear the criteria driving the patients enrollment as well as the precise time of sampling for each patient to obtain 228 specimens. Also, the Authors should describe the evaluation of the specificity of the test. As They use an Ag-RDT detecting also RSV I suggest to add a little background about this virus in the introduction section. Finally, I suggest to discuss more in depth the more usefulness of the anterior nasal specimens compared to the nasopharyngeal based on their good results.

Minor comments

1)Line 176-177: I suggest to better specify 114 nasopharyngeal and 114 anterior nasal specimens.

2)Table 1: likely for me, but I don’t understand why only 97 samples were analysed with RapidTesta

SARS-CoV-2.

3)Figure 3 is not available to be visualized in the pdf file.

4)Figure 1, 2 and 3 legends should be rewrite without results sentences.

Author Response

To Reviewer 3,

Major comments

The article is well written and interesting. The statements defined are supported by detailed presented data. In my opinion, a weakness of this work is represented by the description of sampling that reports detail about the disease but it is not clear the criteria driving the patients enrollment as well as the precise time of sampling for each patient to obtain 228 specimens. Also, the Authors should describe the evaluation of the specificity of the test. As They use an Ag-RDT detecting also RSV I suggest to add a little background about this virus in the introduction section. Finally, I suggest to discuss more in depth the more usefulness of the anterior nasal specimens compared to the nasopharyngeal based on their good results.

Response: Thank you for taking the time to review our manuscript and your pertinent questions and comments.

In my opinion, a weakness of this work is represented by the description of sampling that reports detail about the disease but it is not clear the criteria driving the patients enrollment as well as the precise time of sampling for each patient to obtain 228 specimens.

Response: Regarding the criteria for enrolling patients in our study, all patients who were admitted to Jichi Children’s Medical Center in Tochigi and the Intensive Care Unit and Emergency Center of Jichi Medical University Hospital with suspected infection by SARS-CoV-2 or RSV during the two-month period from February 1, 2022, to March 31, 2022, were included. Eligible patients were checked for SARS-CoV-2 or RSV antigens; if they did not meet the exclusion criteria, the study was explained to them, and if they or their guardians gave written consent, they were included in the study. A new flowchart was created as the flow of patient inclusion and is shown in Figure 1. In addition, its contents were added as follows.

Materials and Methods, Lines 135-141

All patients admitted with suspected SARS-CoV-2 or RSV infection were included. Eligible patients were checked for SARS-CoV-2 antigen or RSV antigen by RT-PCR in the hospital laboratory while the attending physician performed various tests for differential diagnosis during routine medical care; patients were excluded if other pathogens were detected or other diagnoses were made. Patients who did not meet the exclusion criteria for this study were then given a written explanation of the study, and those who gave their consent were invited to participate.

Materials and Methods, Lines 146-150

The exclusion criteria for this study were as follows: (i) high risk of bleeding from the nasal site due to concurrent bleeding disorders, (ii) inability or unwillingness of the patients or their guardians to provide informed consent, (iii) the date of first specimen collection was already > 10 days after disease onset.

Regarding the exact time of sampling for each patient, we felt it was necessary to minimize the stress of specimen collection on the patients as much as possible when sampling over time. Therefore, instead of sampling at a fixed time, we sampled after the patients woke up from sleep, before eating or feeding, before the nurse performed oral or nasal suctioning, or before the physical therapist performed rehabilitation, so as not to interfere with the patient's treatment and recuperation. The following text has been inserted.

Materials and Methods, Lines 175-177

Specimens were collected before the patient ate food, after the patient woke up from sleep, and before the nurse or physical therapist performed oral care or rehabilitation, so as not to interfere with the patient's treatment or recuperation.

the Authors should describe the evaluation of the specificity of the test.

Response: Thank you for pointing this out. The patients with COVID-19 who participated in this study were only moderately to severely ill; asymptomatic and mildly ill patients were unavailable, and hence, not included in the study. Therefore, the results of the analysis of nasal virus dynamics in this study focused on the transition of patients with moderate to severe disease and do not fully reflect that of asymptomatic and mildly ill patients. Furthermore, the number of SARS-CoV-2 antigen-negative specimens was low. Fewer antigen-negative specimens may lead to an underestimation of the nonspecific response of the Ag-RDT and, consequently, to a higher specificity. In summary, a limitation of this study is the clinical predominance of patients with moderate to severe disease, the lack of asymptomatic or mildly symptomatic patients, and the small number of antigen-negative patients. To improve clarity, we have added the following text to the limitation section of the text.

Discussion, Lines 467–470

This study has three limitations. The participants in this study were mainly adults and children with moderate to severe disease, and were few in number. Therefore, our analysis of nasal viral dynamics does not reflect the viral dynamics of all Omicron variant-infected patients, including asymptomatic and mildly ill patients.

In addition, we made the following changes to the text describing the nasal viral dynamics in the Results and Discussion sections to avoid any misunderstanding by the reader.

Results, Lines 248–250

Of all 228 specimens collected mainly from patients with moderate to severe disease, the number of specimens positive for SARS-CoV-2 by RT-PCR was 111 for anterior nasal specimens and 111 for nasopharyngeal specimens.

Discussion, Lines 407–409

In the viral dynamics of the Omicron variant, mainly in patients with moderate to severe disease, the Ct values of the anterior nasal specimens remained close to those of the nasopharyngeal specimens from early onset to day 15 of disease onset.

On the other hand, the specimens we analyzed were always collected from the nasal vestibular and nasopharyngeal sites at the same time, for a total of 228 specimens. Furthermore, since we collected specimens from the anterior nasal and nasopharyngeal sites at different times after disease onset, we were able to collect specimens with various virus levels. Therefore, although the small number of patients is a limitation, we were able to analyze a large number of specimens containing virus amounts just below the detection limit for Ag-RDT evaluation, and we believe that we were able to accurately evaluate the sensitivity of the Ag-RDT.

As They use an Ag-RDT detecting also RSV I suggest to add a little background about this virus in the introduction section.

Response: Thank you for your suggestion. Per your suggestion, we have added the RSV infection symptoms and disease burden, as well as the need for accurate diagnosis, in the Introduction section.

Introduction, Lines 65-78

Respiratory syncytial virus (RSV), like other respiratory viruses such as parainfluenza virus and human metapneumovirus, causes acute upper and lower respiratory tract infections (bronchitis, bronchiolitis and pneumonia). Notably, in infants, transferable maternal antibodies are ineffective in protecting against infection, and a high rate of bronchitis and pneumonia are associated with respiratory failure at the time of initial infection. In fact, about half of infants infected with RSV have lower respiratory tract infections, and 3% of all infants are hospitalized [9]. Lower respiratory tract infections caused by RSV in older individuals are also known to be complicated by wheezing in 6–30% of cases [10]. Furthermore, RSV re-infection exacerbates the chronic diseases in older patients with underlying diseases such as chronic respiratory diseases (asthma and chronic obstructive pulmonary disease). Therefore, early diagnosis and early infection prevention measures are important in homes, daycare centers, and hospitals to avoid transmission to infants and older individuals from those with mild cases of infection.

Introduction, Lines 96-100

As with SARS-CoV-2, RSV diagnosis can be made by PCR, cell culture, and Ag-RDT antigen detection using enzyme antibody or immunochromatography, but the sensitivity of Ag-RDT for detecting RSV antigen is generally around 70% to 80%, and more accurate diagnostic methods are needed [21].

I suggest to discuss more in depth the more usefulness of the anterior nasal specimens compared to the nasopharyngeal based on their good results.

Response: Thank you for your suggestion. We have discussed the usefulness of anterior nasal specimens with new citations. The following text has been added to the Discussion section.

Discussion, Line 407-415

In the viral dynamics of the Omicron variant, mainly in patients with moderate to severe disease, the Ct values of the anterior nasal specimens remained close to those of the nasopharyngeal specimens from early onset to day 15 of disease onset. The Ct values of the anterior nasal and nasopharyngeal specimens of the Omicron variant were not significantly different after day 2 of onset. In other words, viral multiplication in the anterior nasal site was almost equivalent to that at the nasopharyngeal site. This was significantly different from the viral dynamics of the Delta variant over time at the anterior nasal site [24]. In patients with Omicron infection, self-testing using anterior nasal sites is more accurate than that observed for the previously prevalent variants.

Minor comments

  • Line 176-177: I suggest to better specify 114 nasopharyngeal and 114 anterior nasal specimens.

Response: Thank you for your suggestion. We have revised the text to clarify that the nasopharyngeal site is the back part of the nose, and the anterior nasal site is the nostril site, which is the front part of the nose.

Results, Lines 228-230

One hundred and fourteen specimens collected from the anterior nasal site near the nostrils and from the nasopharyngeal site, which is the deep nasal site, were included in the analysis.

  • Table 1: likely for me, but I don’t understand why only 97 samples were analysed with RapidTesta SARS-CoV-2.

Response: Thank you for your question. After a visual determination with RapidTesta SARS-CoV-2, 14 specimens were not analyzed in RapidTesta Reader. Therefore, we have added the following text as a Footnote in Table 2 to avoid any misunderstanding among readers.

Table 2, Footnote,

After the visual evaluation with RapidTesta SARS-CoV-2 was completed, 14 specimens were not analyzed with the RapidTesta Reader. Therefore, the total number of specimens in the RapidTesta Reader was 97.

  • Figure 3 is not available to be visualized in the pdf file.

Response: We apologize for this error. We will ensure that it is visible when resubmitting.

  • Figure 1, 2 and 3 legends should be rewrite without results sentences.

Response: Thank you very much for your suggestion. We have revised the text as follows:

Figure 2: Comparison of severe acute respiratory syndrome coronavirus two-cycle threshold (Ct) values. Mean Ct values are indicated by horizontal bars.

Figure 3A: The cycle threshold (Ct) over time in patients aged ≥ 7 y with coronavirus disease.

Figure 3B: Comparison of cycle thresholds (Ct) over time in patients aged ≤ 6 years with coronavirus disease.

Figure 4; Comparison of the respiratory syncytial virus cycle threshold (Ct) values.

Reviewer 4 Report

Comments and Suggestions for Authors

hi

I found this paper intersting but needs some critical revision

I think abstract should be revised and better explain the rational of the study and its design. I found it difficult to understand and not atractive enough

introduction section should include also clinical signifiance of omicron infection, description of the reason to diagnose it and a comparison to the previous strains

methods- should include flowchart or figure that will explain the design 

I think study population is very problematic- only six pediatric and four adult were included ; some were vaccinated and some not - its hetrogenic and you must treat it properly: maybe add analysis for children separate from vaccinated adults ? 

a comparison of vaccnated vs non-vaccinated ? please consult a statystician in order to overcome this major limitation of the study 

I dont think you can conclude from a study including 10 patients only with wide range of age -  0 months and elderly patients as you evaluated - you should mention it the abstract and discussion - its only proof of concept 

anyway table 1 with clinical and demographic data should be included 

how do you explain RSV positivity in 50% of the patients? is it a false positive ? true finding? superinfection? 

Figure 3 is not clear

please add study sample as a limitation of the study 

please add clinical implications and novelty paragrphs to the discussion 

Comments on the Quality of English Language

Minor editing of English language required

Author Response

To Reviewer 4,

-I think abstract should be revised and better explain the rational of the study and its design. I found it difficult to understand and not attractive enough

Response: Thank you for reviewing our manuscript. We have rewritten the entire abstract to improve its clarity and to help readers understand the purpose and rationale of our study, as suggested.

-introduction section should include also clinical significance of omicron infection, description of the reason to diagnose it and a comparison to the previous strains

Response: Thank you for your suggestion. We have made significant changes from Line 50 in the Introduction, accordingly. This includes a comparison with the Delta variant, the need for diagnosis, clinical importance, and new findings.

-methods- should include flowchart or figure that will explain the design

Response: Regarding the criteria for enrolling patients in our study, all patients who were admitted to Jichi Children’s Medical Center in Tochigi and the Intensive Care Unit and Emergency Center of Jichi Medical University Hospital with suspected infection by SARS-CoV-2 or RSV infection during the two-month period from February 1, 2022, to March 31, 2022, were included. Eligible patients were checked for SARS-CoV-2 or RSV antigens; if they did not meet the exclusion criteria, the study was explained to them, and if they or their guardians gave written consent, they were included in the study. A new flowchart was created as the flow of patient inclusion and is shown in Figure 1. In addition, its contents were added as follows:

Materials and Methods, Lines 135-141

All patients admitted with suspected SARS-CoV-2 or RSV infection were included. Eligible patients were checked for SARS-CoV-2 antigen or RSV antigen by RT-PCR in the hospital laboratory while the attending physician performed various tests for differential diagnosis during routine medical care; patients were excluded if other pathogens were detected or other diagnoses were made. Patients who did not meet the exclusion criteria for this study were then given a written explanation of the study, and those who gave their consent were invited to participate.

Materials and Methods, Lines 146-150

The exclusion criteria for this study were as follows: (i) high risk of bleeding from the nasal site due to concurrent bleeding disorders, (ii) inability or unwillingness of the patients or their guardians to provide informed consent, (iii) the date of first specimen collection was already > 10 days after disease onset.

-I think study population is very problematic- only six pediatric and four adult were included ; some were vaccinated and some not - its hetrogenic and you must treat it properly: maybe add analysis for children separate from vaccinated adults ? a comparison of vaccnated vs non-vaccinated ? please consult a statistician in order to overcome this major limitation of the study

Response: Thank you for your suggestion. Accordingly, we discussed with two statisticians regarding the groups to be compared when performing a significant difference test for viral shedding. For the tests by age and vaccination status, we concluded that it was not possible to divide the cases into several groups because of the limited number of cases. In addition, there were more than 200 specimens collected on different days after disease onset, but since multiple specimens were collected from the same patient, we concluded that these specimens were also unsuitable for statistical examination.

-I don’t think you can conclude from a study including 10 patients only with wide range of age - 0 months and elderly patients as you evaluated - you should mention it the abstract and discussion

Response: Thank you for pointing this out. The patients with COVID-19 who participated in this study were only moderately to severely ill; asymptomatic and mildly ill patients were unavailable, and hence, not included in the study. Therefore, the results of the analysis of nasal virus dynamics in this study focused on the transition of patients with moderate to severe disease and do not fully reflect that of asymptomatic and mildly ill patients. Furthermore, the number of SARS-CoV-2 antigen-negative specimens was low. Fewer antigen-negative specimens may lead to an underestimation of the nonspecific response of the Ag-RDT and, consequently, to a higher specificity. In summary, a limitation of this study is the clinical predominance of patients with moderate to severe disease, the lack of asymptomatic or mildly symptomatic patients, and the small number of antigen-negative patients. To improve clarity, we have added the following text to the limitation section of the text.

Abstract, Line 35-38

The study included symptomatic patients who required hospitalization for oxygen and ventilator management. Two specimens each were collected from the anterior nasal and nasopharyngeal sites, with a total of 228 specimens from 10 patients.

Discussion, Lines 467–470

This study has three limitations. The participants in this study were mainly adults and children with moderate to severe disease, and were few in number. Therefore, our analysis of nasal viral dynamics does not reflect the viral dynamics of all Omicron variant-infected patients, including asymptomatic and mildly ill patients.

In addition, we made the following changes to the text describing the nasal viral dynamics in the Results and Discussion sections to avoid any misunderstanding by the reader.

Results, Lines 248–250

Of all 228 specimens collected mainly from patients with moderate to severe disease, the number of specimens positive for SARS-CoV-2 by RT-PCR was 111 for anterior nasal specimens and 111 for nasopharyngeal specimens.

Discussion, Lines 407–409

In the viral dynamics of the Omicron variant, mainly in patients with moderate to severe disease, the Ct values of the anterior nasal specimens remained close to those of the nasopharyngeal specimens from early onset to day 15 of disease onset.

- its only proof of concept

Response: Thank you for pointing this out.

Although the participants in our study were few in number, the number of specimens collected is 218, which includes specimens with various Ct values. Therefore, we believe that the sensitivity evaluation of Ag-RDT has been demonstrated in our study.

-anyway table 1 with clinical and demographic data should be included

Response: Thank you for your suggestion.

Accordingly, we have added a list as Table 1 that includes the detailed clinical presentation and treatment of the patient.

-how do you explain RSV positivity in 50% of the patients? is it a false positive? true finding? superinfection?

Response: Thank you for your question. We apologize for the confusing wording. First, all 228 specimens were evaluated using RapidTesta RSV & SARS-CoV-2, and there were no RS antigen positive specimens. However, when we used RT-PCR to detect RSV antigen, we determined that five of the anterior nasal specimens and nine of the nasopharyngeal specimens were positive for RSV antigen. In other words, all RSV antigen-positive specimens were false negative by RapidTesta RSV & SARS-CoV-2. For the detection of RSV antigens in RapidTesta RSV & SARS-CoV-2, we have added a technical modification to RapidTesta RSV-Adeno NEXT, which is already in clinical use. The diagnostic accuracy of this RapidTesta RSV-Adeno NEXT for RSV antigen has been confirmed to be 98.1 in sensitivity and 100% in specificity when the RT-PCR method is used as the gold standard. The reason for the false negative result for RSV antigen in RapidTesta RSV & SARS-CoV-2 this time was the low amount of RSV, i.e., the specimen had a high Ct value, but the apparent cause is unknown.

For reader clarity and to avoid misunderstanding, the text within the RSV positively section has been changed as follows.

Results, Line 349-353

A visual inspection and RapidTesta Reader determination of the 228 specimens showed negative results for RSV. However, RT-PCR showed positive results for RSV in five anterior nasal specimens and nine nasopharyngeal specimens. In a subtype analysis of viral pathogens, all specimens detected as RSV were of type A and showed co-infection with SARS-CoV-2.

-Figure 3 is not clear

Response: We apologize for this and will ensure visibility when resubmitting.

-please add study sample as a limitation of the study

Response: Thank you for your suggestion. A limitation of this study is the lack of asymptomatic or mildly symptomatic patients, with a clinical focus on patients with moderate to severe disease. We have added the following text to the limitation section of the manuscript to make it easier for readers to understand.

Discussion, Line 467-470

The participants in this study were mainly adults and children with moderate to severe disease, and were few in number. Therefore, our analysis of nasal viral dynamics does not reflect the viral dynamics of all Omicron variant-infected patients, including asymptomatic and mildly ill patients.

-please add clinical implications and novelty paragrphs to the discussion.

Response: Thank you for your suggestion.

Many Ag-RDTs are currently in clinical use, and SARS-CoV-2 is a fast-mutating virus, with various variants ranging from the Alpha to Delta to Omicron variants occurring in the 3 years since the pandemic. It is unclear whether the various S protein variants affect the diagnostic efficacy of Ag-RDT. The emergence of new variants and the evaluation of the accuracy of the newly developed Ag-RDTs should be done aggressively with clinical specimens. In this context, the newly developed Ag-RDTs evaluated in this study was highly accurate in diagnosing the Omicron variants. Furthermore, we found that anterior nasal specimens were more useful for detecting Omicron variants than nasopharyngeal specimens. The usefulness of anterior nasal specimens in future clinical practice has been discussed in depth. The following text has been newly added to the Discussion section.

Discussion, Lines 423-434

The Ct values for Omicron at the anterior nasal site in patients with moderate to severe were greatly reduced compared to those for Delta [24], suggesting high proliferation of Omicron at the anterior nasal site. Changes in the Omicron viral levels in the nasal cavity were not evident. Omicron infection has been shown to result in an increased dependence on cathepsin B instead of dependence on transmembrane serine protease 2 (TMPRSS2) [32], which is very different from Delta infection. We speculate that it may be because there is higher cathepsin B expression than TMPRSS2 expression at the anterior nasal site. A high viral load at the anterior nasal site up to 5 days after disease onset has been suggested to lead to the spread of infection [8]; however, our results, analyzed in more detail, revealed that the viral load at the anterior nasal site increased to the same level as that at the nasopharyngeal site from early after onset. This may be involved in the spread of Omicron infection.

Discussion, Line 450-453

The Ag-RDTs we evaluated in this study proved to have high diagnostic accuracy even for the Omicron variant, which has various mutations in the viral surface spike protein structure and is currently the overwhelming mainstream of the epidemic.

Reviewer 5 Report

Comments and Suggestions for Authors

The topic is significant and relevant to the reader's interest. However, there are some issues required to be addressed. 

Introduction: 

1. Further literature review is required on the available diagnostic tests for SARS-CoV-2 Omicron variants. 

2. Please highlight the objectives of the study as a separate paragraph. 

Materials and Methods:

Please add a figure on the methods and process of data collection.

Author Response

To Reviewer 5,

The topic is significant and relevant to the reader's interest. However, there are some issues required to be addressed.

-Introduction:

  1. Further literature review is required on the available diagnostic tests for SARS-CoV-2 Omicron variants.

Response: Thank you for your suggestion. We have added the following information to the Introduction section, including the reaction mechanism of the Ag-RDT with viruses and the usefulness of Ag-RDT for detecting the Omicron variant.

Introduction, Lines 93-100

Many specimens are licensed for use, including nasopharyngeal and pharyngeal site specimens, followed by saliva, nasopharyngeal aspirate, and sputum, although those used for self-testing are often limited to saliva or anterior nasal site specimens [17,18]. Studies on the detection efficacy of the SARS-CoV-2 variants for Ag-RDT are divided [19,20]. As with SARS-CoV-2, the diagnosis of RSV can be made by PCR, cell culture, and Ag-RDT antigen detection using enzyme antibody or immunochromatography, but the sensitivity of Ag-RDT for detecting RSV antigen is generally around 70% to 80%, and more accurate diagnostic methods are needed [21].

  1. Please highlight the objectives of the study as a separate paragraph.

Response: Thank you for your suggestion. We have added the following sentence as a separate paragraph in the Introduction section.

Introduction, Line 117-126

The objective of our study was to first determine the nasal viral dynamics of the Omicron variant. We analyzed the viral load and its excretion-course over time in two nasal sites, one in the anterior nasal site and the other in the nasopharyngeal site. The Omicron variant viral titer in the anterior nasal site was used to discuss whether it could be applied toward self-testing. Another objective of the study was to validate the diagnostic ability of the newly developed Ag-RD, which is a combination of separate Ag-RDTs already in use to identify SARS-CoV-2 antigen and RSV antigen within a single Ag-RDT. The basic principle of the detection of SARS-CoV-2 and RSV antigens is based on existing technology with many technical improvements. The diagnostic accuracy of the newly developed Ag-RDT was evaluated using clinical specimens, as its diagnostic accuracy was not clear.

-Materials and Methods:

Please add a figure on the methods and process of data collection.

Response: Thank you for your suggestions. We have now created a flowchart of the data collection method and process, which is newly designated as Figure 1.

Round 2

Reviewer 1 Report

Comments and Suggestions for Authors

Although numerous changes have been made, not all answers were satisfactory. In particular, including asymptomatic and paucisymptomatic in the analyzed population should be mandatory. The selection of the population represents a clear bias for the validation of this Ag-RDT.

Comments on the Quality of English Language

Minor editing of English language required

Reviewer 2 Report

Comments and Suggestions for Authors

Dear authors

Thanks for this manuscript

Reviewer 4 Report

Comments and Suggestions for Authors

authors have addressed my comments